# Identifying relevant common sense information in knowledge graphs

**Guy Aglionby** and **Simone Teufel**
Department of Computer Science and Technology
University of Cambridge
United Kingdom
{guy.aglionby,sht25}@cl.cam.ac.uk

## Abstract

Knowledge graphs are often used to store common sense information that is useful for various tasks. However, the extraction of contextually-relevant knowledge is an unsolved problem, and current approaches are relatively simple. Here we introduce a triple selection method based on a ranking model and find that it improves question answering accuracy over existing methods. We additionally investigate methods to ensure that extracted triples form a connected graph. Graph connectivity is important for model interpretability, as paths are frequently used as explanations for the reasoning that connects question and answer.

## 1 Introduction

For models to be able to reason about situations that arise in everyday life, they must have access to contextually appropriate common sense information. This information is commonly stored as a large set of facts from which the model must identify a relevant subset. One approach to structuring these facts is as a knowledge graph. Here, nodes represent high-level concepts, and typed edges represent different kinds of relationship between concepts. In practice, a subset of facts that are thought to be contextually relevant are extracted from the graph, as using all facts in each instance is unnecessary, noisy, and computationally expensive.

Prior work has focused on different ways to encode these facts, including by inputting them into a graph neural network (GNN) or into a transformer (Feng et al., 2020; Yasunaga et al., 2021). However, the question of how to identify useful information has been under-explored, particularly in work that uses GNN encoders. If contextually important information is not retrieved then performance could be dramatically reduced, a potential result of the use of overly simplistic retrieval methods.

In this paper we explore methods to extract high-quality subgraphs containing contextually relevant

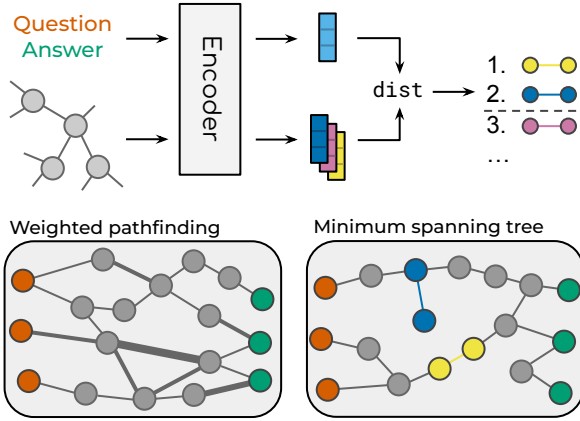

Figure 1: The triple scoring process for a question answering task, and two methods that use the scores to extract relevant subgraphs for a question and candidate answer.

information.[1] We approach this as a ranking task across triples in a knowledge graph, and propose two methods that use the scores to extract a subgraph. The first is a weighted pathfinding approach which extends prior work (Lin et al., 2019), while the second builds a minimum spanning tree that includes the highest-ranked triples (figure 1). Both approaches ensure that all or most nodes in the subgraph are reachable from each other, which is important for two reasons. First, it means that the GNN can update node embeddings with information from most other nodes, which would not be possible if the graph were disconnected. Second, it allows paths of reasoning to be extracted from the subgraph, which are often used as explanations for model behaviour (Feng et al., 2020; Wang et al., 2020; Yasunaga et al., 2021).

There are also situations when specific concepts need to be included in order for a subgraph to be of high enough quality. For example, in question answering, a full explanation must include one

---

[1]We call these "relevant subgraphs" or "extracted subgraphs", noting that others use "schema graphs" (Lin et al., 2019).

or more concepts mentioned both in the question and in a candidate answer. This requires robustness towards how concepts identified because the knowledge repository might express the concept in a slightly different lexical form from the question and/or answer. We therefore experiment with a embedding-based method to identify these concepts, and compare it with existing lexical methods.

Our contributions are as follows[2]:

- Apply a ranking model to identify common sense triples that are relevant to some context.
- Identify and thoroughly investigate methods to ensure that the extracted contextually-relevant subgraphs are (almost) connected.
- Compare existing lexical approaches to entity linking to a simple embedding-based method.

## 2 Background

Many prior approaches to retrieving relevant common sense triples from a knowledge graph start by identifying relevant nodes. Simple lexical overlap between a concept and the context (e.g. question text) is often used for this (Kundu et al., 2019; Khot et al., 2019). However, this entity linking approach is likely to only retrieve simple concepts, as the idiosyncratic phrasing of some node names in knowledge graphs like ConceptNet (Speer et al., 2017) are unlikely to show up in text. Becker et al. (2021) investigate this in detail and propose a series of pre-processing steps that allow lexically-based linking without exact phrase matches. For the same reason, the heuristics used by Lin et al. (2019) for lexical matching are employed by a series of later works (Feng et al., 2020; Yasunaga et al., 2021; Wang et al., 2020). Although lexical matching is a frequent approach with common sense knowledge graphs, in other domains embedding-based approaches are more popular (Gillick et al., 2019). These work by embedding the candidate text and finding the nearest neighbour in the space of entity embeddings.

In question answering, Lin et al. (2019) split these concepts into those identified in the question and in the answer, and find additional concepts for the relevant subgraph by iteratively finding shortest paths between the two sets. This process continues until a maximum number is collected, or the path lengths exceed a threshold. The final subgraph used as input to models is constructed from this set with all valid edges added.

Some approaches score nodes and triples that have been identified. Kundu et al. (2019) score multiple paths for each question and answer and choose the answer with the highest mean path score. Yasunaga et al. (2021) extract a subgraph following Lin et al. (2019), and additionally score each node for relevance to a question using RoBERTa (Liu et al., 2019). Ranking is also common with prose facts, particularly when they are input into transformer-based models that have limits on input size (Wang et al., 2021).

## 3 Methodology

In this section we introduce our methods for extracting a contextually-relevant subgraph $\mathcal{G}$ for a question answering task. The graph should contain triples that are useful in distinguishing the correct answer from a set of distractors. For each instance, we represent the question text as $q$ and the $i$th candidate answer as $a_i$, and the set of concepts extracted from each as $\mathcal{C}_q$ and $\mathcal{C}_{a_i}$ respectively.

### 3.1 Triple scoring

We cast the task of identifying relevant triples in the knowledge graph as a ranking problem, where the highest-ranked triples are those most relevant to $q; a_i$. We use an existing model that is trained to rank facts highly if they constitute part of an explanation for why $a_i$ is the correct answer to $q$ (Pan et al., 2021). This was developed for the TextGraphs 2021 shared task on explanation regeneration for science questions (Thayaparan et al., 2021) and achieved the highest performance. Facts that are used in an explanation are likely to be useful when choosing between answers, making the model a natural choice for identifying relevant triples.

The model consists of two parts: a fact retriever and a re-ranker. We follow the training procedure in Pan et al. (2021) and use one model based on RoBERTa-Large (Liu et al., 2019) for each stage. At inference time we use only the re-ranker to score each triple[3] in relation to $q; a_i$. To speed this up we pre-compute embeddings for each $q; a_i$ and each triple.

---

[2]We make our code and data available at https://github.com/GuyAglionby/kg-common-sense-extraction.

[3]We linearize triples using the templates from https://github.com/commonsense/conceptnet5/wiki/Relations.

## 3.2 Constructing $\mathcal{G}$

The most straightforward way to construct $\mathcal{G}$ is to use the most relevant triples identified in §3.1 and the grounded nodes $\mathcal{C}_q \cup \mathcal{C}_{a_i}$. To do this, we select a subset of the top $e$ ranked triples according to limits on the total number of edges and nodes that would be added to $\mathcal{G}$. Iterating in rank order, we add the triple $(s, r, o)$ to $\mathcal{G}$ only if adding $s$ and $o$ does not increase the total number of nodes to above $n$. If $n < 2e$ then some of the top edges will be excluded; this limits the number of nodes in the graph while allowing highly-ranked edges to be present if they share nodes with other edges. We set $n = 50$ and $e = 40$ following initial experiments.

A shortcoming of this method is that the selected triples are not likely to connect with $\mathcal{C}_q$ or $\mathcal{C}_{a_i}$. Indeed, there is no guarantee that the triples are connected to each other. This is problematic in cases where paths in the extracted subgraph are to be used in an explanation (Feng et al., 2020; Yasunaga et al., 2021).

To rectify this we find the minimum spanning tree (MST) that spans all nodes in $\mathcal{G}$, taking into account the edges added in the previous step. This is the Steiner tree problem, which is NP-hard; we apply an approximation algorithm (Wu et al., 1986) to find solutions in a reasonable amount of time. We experiment with two variants: one where edges are uniformly weighted, and another where the triple scores are used as weights.

We further use the triple scores with the pathfinding method used in previous work (Lin et al., 2019), transforming this into a weighted shortest path search. We iteratively find the shortest path between any pair of concepts in $\mathcal{C}_q$ and $\mathcal{C}_{a_i}$, adding nodes on the paths to a set until a maximum size is reached. $\mathcal{G}$ is then formed from these nodes, as well as all valid edges between pairs from this set. We set the maximum number of nodes to be 50.

## 3.3 Identifying relevant concepts

It is important that $\mathcal{C}_q$ and $\mathcal{C}_{a_i}$ accurately reflect concepts mentioned in $q$ and $a$, primarily to aid with explanations. A full explanation for a question must include at least one concept from $\mathcal{C}_q$ and from $\mathcal{C}_{a_i}$; if these concepts are nonsensical then the explanation is invalid. Additionally, the pathfinding method for relevant subgraph extraction relies on the quality of this grounding.

We use two methods for entity linking. The first is from prior work, and is based on lexical match-

ing with heuristics (Lin et al., 2019). These include lemmatising words if an exact match is not found, and a method to avoid selecting nodes with lexical overlap. Despite this, lexical methods are not able to identify relevant concepts that have a lexical form that is not likely to be seen in any context; this occurs often with more specific concepts. To account for this, our second method is based on embeddings from `RoBERTa`. We embed each concept, and for each $q$ and $a_i$ find the 10 most similar concepts via Euclidean distance. Embeddings are constructed in each case by mean-pooling across all tokens.

## 3.4 Evaluation

We evaluate the quality of the extracted subgraphs by comparing accuracy on a question answering task when using them versus using a baseline. These graphs are used as input to two models, MH-GRN (Feng et al., 2020) and QA-GNN (Yasunaga et al., 2021), which are both designed for question answering with knowledge graphs. The baseline subgraph is extracted using the unweighted pathfinding method from prior work (Lin et al., 2019); for the fairest comparison we run five baselines which extract subgraphs of different sizes and report the best result from these (see appendix C for full details). We also compare to baseline that uses only `RoBERTa-large` with no additional facts.

We report accuracy on two datasets, OpenbookQA (Mihaylov et al., 2018) and CommonsenseQA (Talmor et al., 2019). OpenbookQA is a collection of science questions, and so is in-domain with respect to the data used to train the fact scorer. CommonsenseQA targets more general common sense; performance here is a reflection on how transferable the fact scorer is to other domains. This dataset has no public test set labels, so we report results on the 'in house' test split defined by Lin et al. (2019). Each model is run three times with different random seeds and the mean accuracy reported. Model hyperparameters are reported in appendix A.

Our base knowledge graph is ConceptNet (Speer et al., 2017). Following previous work (Lin et al., 2019), we merge similar relations and add reverse relations to the extracted graph.

| Grounding | Subgraph type | MHGRN | QA-GNN |
|---|---|---|---|
| *LM Only* | | 62.07 | |
| Lexical | *Baseline* | **67.73** | **67.07** |
| | Only top rated | 62.73 | 64.47 |
| Lexical | MST | 63.07 | 64.27 |
| | Weighted MST | 64.87 | 60.73 |
| | Weighted path | 64.20 | 65.27 |
| Embedding | MST | 65.47 | 66.33 |
| | Weighted MST | 64.73 | 64.60 |
| | Weighted path | 64.07 | 65.73 |

Table 1: Accuracy on **OpenbookQA** with different subgraph extraction methods.

| Grounding | Subgraph type | MHGRN | QA-GNN |
|---|---|---|---|
| *LM Only* | | 69.53 | |
| Lexical | *Baseline* | 69.48 | 70.32 |
| | Only top rated | 69.76 | 69.92 |
| Lexical | MST | 69.86* | 69.35 |
| | Weighted MST | 69.19 | **70.64*** |
| | Weighted path | 69.86* | 68.87 |
| Embedding | MST | 69.60 | 70.10 |
| | Weighted MST | **69.97*** | 69.86 |
| | Weighted path | 69.27 | 70.08 |

Table 2: Accuracy on **CommonsenseQA** with different subgraph extraction methods.[5]

## 4 Results

Our results on OpenbookQA are presented in table 1 and CommonsenseQA in table 2. On CommonsenseQA, our best method significantly[4] outperforms the baseline method. This suggests that, in this case, the ranker is able to identify facts which are relevant to the question, and that the models are subsequently able to successfully use them.

The tuned baseline for OpenbookQA beats the proposed methods in all cases, although there is reasonable variation in accuracy between the baselines of different sizes (see table 6). However, in all but two cases the methods for ensuring graph connectivity outperform the method that only uses the highest-ranked triples.

## 5 Analysis

We observe that, in the majority of cases, using methods to increase connectivity within the extracted subgraph improves performance over simply including the top rated facts. The minimum spanning tree (MST) approach has the advantage of including these facts, unlike the weighted path method which may not. However, to ensure that the graph is connected the MST approach may have to include nodes and edges that are less relevant to the context. One might expect a weighted approach to counterbalance this, however this also results in a larger subgraph being constructed which may be detrimental (see appendix B). Indeed, with lexical grounding the weighted approach adds an average of 37 nodes and 83 edges to the extracted subgraph, compared with 26 nodes and 71 edges in the unweighted case.

The weighted pathfinding approach has the advantage of avoiding edges which are not relevant to the query. Additionally, the subgraph is extracted in way that is closer to $\mathcal{C}_q$ and $\mathcal{C}_{a_i}$ than the MST approach, which considers these nodes only after selecting the top-ranked triples. As a result, the question and answer nodes are connected in a larger variety of ways, which may help increase performance.

For OpenbookQA, the increase in score between lexical and embedding-based entity linking with an unweighted MST suggests that the concepts identified by the latter method are particularly useful. The same magnitude of increase is not seen in CommonsenseQA. One possible reason for this is that CommonsenseQA was constructed directly using ConceptNet, which may increase the relevance of concepts obtained with lexical methods.

Similarly to with lexical grounding, the weighted MST with embedding grounding adds more nodes and edges on average (153 nodes, 217 edges) than the unweighted one (112 nodes, 172 edges). In both cases, the resulting subgraph is substantially larger than the equivalent ones built from lexically-linked entities. This is likely due to the kinds of nodes identified by entity linking – we observe that concepts identified by the embedding-based method are more specific, and so are less connected within the overall graph. Conversely, concepts that are identified lexically are likely to be simpler and more general, and so better connected within the graph, meaning fewer additional nodes and edges are required to build the MST.

---

[4] We use the Almost Stochastic Dominance test (Dror et al., 2019) and only claim a significant difference if $\epsilon \leq 0.05$.

[5] * denotes significantly better than baseline subgraph at $p < 0.001$.

## 6 Conclusion

We present a method for extracting relevant information from a common sense knowledge graph, casting it as a ranking problem. We show that scores obtained from a ranking model can be used to select triples containing useful information for a question answering task, improving performance over a commonly-used approach.

As it is undesirable for extracted subgraphs to have low connectivity, particularly when using paths within them for model interpretation, we use an algorithm for calculating minimum spanning trees over a supplied set of nodes and edges to ensure the graph is connected. We find that this helps performance; in particular, the models with highest accuracy on CommonsenseQA use a weighted version of this. We additionally find that using an entity linking approach that uses embeddings rather than lexical matching improves performance in some cases. We distribute the contextually-relevant subgraphs to facilitate future work; these drop in to existing models with no further processing required.

Future work might investigate the influence of the fact ranker, as our results suggest that it can transfer from the science to general common sense domain successfully. Further training of the ranker using higher-quality negative samples from e-QASC (Jhamtani and Clark, 2020) may yield better performance, as noted by Pan et al. (2021).

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

## A  Hyperparameters

We use the same hyperparameters for MHGRN and QA-GNN as used in the papers which respectively introduced them (Feng et al., 2020; Yasunaga et al., 2021). We optimise both models using RAdam (Liu et al., 2021) and a learning rate of $1e-3$ for the text encoder and $1e-5$ for the graph encoder. A maximum of 128 tokens are input to the text encoder, which is initialised as RoBERTa-large. A L2 weight decay of 0.01 is used.

For MHGRN, batch size is 32 and the text encoder is frozen for the first 3 epochs. A 1-layer 100-dimensional GNN is used with 3-hop message passing at each layer.

For QA-GNN, batch size is 128 and the text encoder is frozen for the first 4 epochs. A 5-layer 200-dimensional GNN is used.

In all cases, the GNN is initialised with node embeddings derived from BERT, which are made available by Feng et al. (2020).

## B  Extracted subgraph size

For each type of extracted subgraph, we report the mean and standard deviation of the number of edges in table 3 and number of nodes in table 4. We report results for the baselines in table 5.

| Grounding | Subgraph type | OBQA | CSQA |
|---|---|---|---|
| Lexical | Only top rated | 33±6 | 28±5 |
| | MST | 104±28 | 110±29 |
| | Weighted MST | 117±30 | 123±32 |
| | Weighted path | 216±50 | 232±54 |
| Embedding | MST | 202±50 | 201±46 |
| | Weighted MST | 245±64 | 250±56 |
| | Weighted path | 168±43 | 177±47 |

Table 3: Average number of edges in extracted subgraphs for OpenbookQA and CommonsenseQA.

| Grounding | Subgraph type | OBQA | CSQA |
|---|---|---|---|
| Lexical | Only top rated | 49±2 | 50 |
| | MST | 78±22 | 77±21 |
| | Weighted MST | 89±23 | 89±23 |
| | Weighted path | 53±5 | 54±4 |
| Embedding | MST | 167±41 | 162±35 |
| | Weighted MST | 207±53 | 206±45 |
| | Weighted path | 59±3 | 58±2 |

Table 4: Average number of nodes in extracted subgraphs for OpenbookQA and CommonsenseQA.

| Nodes/edges | Model | OBQA | CSQA |
|---|---|---|---|
| Nodes | MHGRN | 50±10 | 36±7 |
| | QA-GNN | 63±12 | 63±12 |
| Edges | MHGRN | 128±23 | 64±13 |
| | QA-GNN | 190±33 | 188±36 |

Table 5: Average number of nodes and edges in baseline subgraphs for OpenbookQA and CommonsenseQA.

| Target edge count | MHGRN | QA-GNN |
|---|---|---|
| 50 | 65.27 | 65.20 |
| 100 | **67.73** | 65.87 |
| 150 | 63.53 | **67.07** |
| 200 | 65.27 | 66.53 |
| 250 | 64.40 | 64.20 |

Table 6: Accuracy on **OpenbookQA** when using the baseline subgraph extraction method with five different target edge counts.

| Target edge count | MHGRN | QA-GNN |
|---|---|---|
| 50 | **69.48** | 70.08 |
| 100 | 68.60 | 69.83 |
| 150 | 69.11 | **70.32** |
| 200 | 68.95 | 69.54 |
| 250 | 69.46 | 69.33 |

Table 7: Accuracy on **CommonsenseQA** when using the baseline subgraph extraction method with five different target edge counts.

## C  Baseline models

Subgraph size is a confounding factor when comparing performance between our extraction methods and the baseline (Lin et al., 2019). To control for this, we extract baseline subgraphs of five different sizes by expanding them until they reach a certain number of edges. In tables 1 and 2 we report the only highest scoring baseline; full baseline results are presented in tables 6 and 7.