# OpenReview forum: "Identifying relevant common sense information in knowledge graphs"
_aclweb.org/ACL/2022/Workshop/CSRR — ACL 2022 Workshop CSRR_

### Official Review · Reviewer_NAhj · 2022-03-15
**Reasonable technical ideas, but lack of analysis, and question mark on reported gains**

**Rating:** 6
**Confidence:** 5

**Review:**

The paper discusses various ways to retrieve CSK statements for QA methods that rely on structured knowledge.

The ideas of zooming in on the retrieval part of the problem, in particular concerning lexical vs. embedding-based retrieval, and graph connectivity, is interesting. The technical content is reasonably presented, and the ideas are worth discussing at the workshop.

At the same time, there are critical issues:
 - Readability for all but a small expert circle is substandard, as the paper is devoid of examples
 - The analysis is narrow, on a single idiosyncratic KG, ConceptNet. The proposed methods would gain much credibility if shown to work not only on that, but also, e.g., on Quasimodo, Atomic, or Ascent.
 - There is no analysis of system outputs, only aggregate numbers. I understand that human evaluations of this task are hard, but I honestly wonder whether the authors ever looked at their data themselves. There should at least be hand-picked comparisons of system outputs. Are there any human insights on how the proposed methods do better (if they do)?
 - The reported gains are small, and it is unclear whether they are statistically significant. Having worked on this setting myself, I am skeptical of suggestions that CSKGs truly help in the problem (and the result tables should show a no-KG baseline). Moreover, there seem to be confounding factors, as different methods produce KGs of different size, which make attributing gains to the proposed methods even harder - perhaps the gains just come from finding the sweet spot of right KG size?

Minor points:
 - The term "schema graph" appears odd. "Schema" to me rather refers to data organization, not to actual grounded triples.
 - "We make our code available at anonymous" - This isn't helpful - there are enough ways nowadays to anonymously share actual files, at a workshop I don't expect code sharing anyway, but at a conference I would not let this pass (if code/data sharing was a requirement).

---

### Official Review · Reviewer_RBxX · 2022-03-19
**Simple, well-written contribution; appropriate fit for workshop**

**Rating:** 8
**Confidence:** 4

**Review:**

The authors introduce a triple selection method based on a ranking model and find that it improves QA accuracy over existing approaches. They also investigate methods to ensure that extracted triples form a connected graph. They argue that this connectivity is important both for model interpretability, and also because non-connectivity could limit the power of the GNN.

Overall, the paper makes a nice contribution to the workshop and I recommend accept. It is relatively simple, and appropriate to its length. In a longer version, I would have liked to see a figure illustrating the intuition behind the approach, but due to page limits this may not have been possible.

It is also always good to include statistical significance results which seems to be missing here.

---

### Official Review · Reviewer_1Bkt · 2022-03-24
**Well written, Clearly communicated all complexities**

**Rating:** 7
**Confidence:** 3

**Review:**

This paper improves contextually relevant knowledge extraction by introducing common sense triple selection based on a ranking model.

[Strong]
Paper is well written with detailed results and analysis demonstrating quality of the extracted schema graphs on two popular models (MHGRN, QA-GNN) and datasets (OpenbookQA, CommonsenseQA). Current papers triple selection based on reranker, iteratively finding the shortest path between any pair of concepts (using steiner tree approximations, pathfinding methods), employing lexical and embeddings based entity linking approaches for extracting schema graphs helps foster further research in knowledge extraction space.

[Weak]
The current approach has lesser incremental gains on CommonsenseQA dataset vs baseline (signifying lesser transferability to other domains) when compared to OpenbookQA dataset vs baseline (signifying higher gains on indomain data)

[Minor Typo]
QA-GNN paper reference is wrongly mentioned in section 3.4

---

### Decision · Program_Chairs · 2022-03-28

Accept